# Viable and Heat-Resistant Microbiota with Probiotic Potential in Fermented and Non-Fermented Tea Leaves and Brews

**DOI:** 10.3390/microorganisms13050964

**Published:** 2025-04-23

**Authors:** Elisabeth Uhlig, Afina Megaelectra, Göran Molin, Åsa Håkansson

**Affiliations:** Department of Process and Life Science Engineering, Lund University, P.O. Box 124, SE-221 00 Lund, Sweden; elisabeth.uhlig@ple.lth.se (E.U.); afina.nuur.farma@gmail.com (A.M.); goran.molin@ple.lth.se (G.M.)

**Keywords:** tea microbiota, Sanger sequencing, tea leaves, tea brew, viable count, *Heyndrickxia coagulans*, fermented tea, potential probiotics

## Abstract

The live microbiota of tea has not been extensively investigated. This study aimed to identify the live, culturable microbiota from four types of tea with varying oxidation levels, before and after brewing. Tea leaves and brews from oolong and fermented teas were analyzed for total viable counts of aerobic bacteria, lactobacilli, fungi, and *Enterobacteriaceae*. Cultivation was performed and isolates were identified by Sanger sequencing. Heat resistance was assessed at 70 °C and 90 °C. Random Amplified Polymorphic DNA (RAPD) was used to determine strain-level diversity. Fully oxidized, post-fermented Pu-erh tea had the highest viable bacterial count. Most isolates belonged to *Bacillaceae*, *Staphylococcaceae,* and *Paenibacillaceae*, families associated with soil or human skin. Only two potentially pathogenic species were identified: *Staphylococcus epidermidis* and *Bacillus cereus*. In Pu-erh, live bacteria were detected after brewing at 90 °C, including *Heyndrickxia coagulans*, a spore forming probiotic species. *H. coagulans* strains remained in vegetative state after hot water exposure and survived at 70 °C, indicating thermotolerance. RAPD-analysis revealed nine distinct *H. coagulans* strains across six Pu-erh teas. Conclusion: This study provides new insight into the viable microbiota of different teas and their survival during brewing, highlighting safety concerns and probiotic species like *H. coagulans*.

## 1. Introduction

As a cultural beverage, tea is, next to water, the most widely consumed beverage in the world [1] and it is well known for its beneficial effects with pronounced antioxidative, antimicrobial and antiviral activity [2,3,4]. Given its global consumption, understanding the microbiological safety and potential health benefits of tea is highly relevant. As interest in functional foods and natural probiotics grows, viable microorganisms in tea may influence both consumer health and innovation in product development. Identifying microbes that survive brewing can also inform public health guidance and support commercial efforts in the expanding global tea market.

Traditionally, tea is made of cured leaves of the evergreen plant *Camellia sinensis* and can be divided into four types according to its processing method: white and green tea, oolong tea, black tea, and dark (post-fermented) tea. In the production of white and green teas, the leaves are disrupted and quickly heat treated after harvest to inactivate oxidation enzymes, resulting in a less oxidized tea [5]. The production of black tea involves withering under sun exposure, disruption, and application of suitable temperature and moisture levels [6] to promote enzymatic oxidation of phenolic compounds. The oxidation process of oolong tea is prematurely disrupted by heat treatment, resulting in only partially oxidized tea. Generally, the degree of oxidation in light oolong tea ranges from 10 to 30%, while in darker oolong tea, the oxidation range is between 60 and 70%. Dark tea, or fermented tea, is post-fermented, mainly by fungi of the genera *Aspergillus* and *Saccharomyces* [5,7]. During the fermentation, the microorganisms will continue the oxidation of phenolic compounds, also referred to as the secondary oxidation, and the fermentation can last for several years [8].

While the chemical components of tea, such as phenolics, are well described [9,10,11,12,13], there is less information about its microbial composition. Zhao et al. [14] analyzed the microbial composition of Pu-erh tea using 454 pyrosequencing and found that the most abundant microorganisms were fungi of the genus *Aspergillus* (95%) and bacteria of the phylum Proteobacteria (48%). Other bacterial phyla included Firmicutes (20%), Actinobacteria (20%), Cyanobacteria (10%), and Bacteriodetes (4%). In other studies, Fu et al. [15] used metagenomics to characterize the microbiota in Pu-erh, and Liu et al. [16] simulated black tea production and found that the bacterial genera *Chryseobacterium* and *Sphingomonas*, and fungi order *Pleosporales* were dominant in the process. These microorganisms significantly influence tea’s quality and potential health benefits. However, limited research has been conducted on their viability, which is crucial for ensuring food safety.

High-throughput sequencing, while powerful, often delivers data at a shallow sequencing depth and cannot differentiate between live and dead microorganisms. The aim of this study was to characterize the viable, culturable microbiota present in three distinct tea types—lightly oxidized oolong, heavily oxidized oolong, and fermented Pu-erh—both before and after brewing. This was achieved through microbial cultivation, identification by Sanger sequencing, and assessment of bacterial heat resistance, with a specific focus on safety aspects and potential probiotic species. Understanding the live microbiota after brewing is essential not only for assessing potential bacteriological risks, but also for gaining deeper insights into the functional potential of tea as a health-promoting beverage. By focusing on the viable fraction of the microbiota, this study addresses both food safety concerns and the growing interest in tea’s functional properties, particularly its probiotic potential.

## 2. Materials and Methods

### 2.1. Sample Collection

The teas subjected to microbiological analysis originated from China and included Tie Guan Yin, a lightly oxidized oolong tea (samples from 2 batches); Jingzhi, a medium oxidized oolong tea (samples from 2 batches); Nóng xiang, a strongly oxidized oolong tea (samples from 2 batches); the dark, post-fermented teas Pu-erh (6 types, 1 batch/type); and Liu Bao Cha (1 batch) purchased from a tea vendor in Sweden (Ehsans & Pappas Tehus, Malmö) and were transported in its original package directly to the laboratory. Liu Bao Cha was purchased directly in China and brought to Sweden in its original package. All samples were collected in 2023. All teas were fresh and not stored for extended periods before being sold. Additionally, all tea samples were stored at ambient temperature in their original packaging prior to analysis, ensuring that the samples were handled according to typical commercial practices. The tea leaf fragments analyzed ranged in size from approximately 0.5 to 1 cm in length and 1 mm in thickness. Six replicates were taken from each batch.

### 2.2. Bacterial Sampling of Tea Brew

Before brewing, 10 g of each kind of tea was sampled by a sterilized spoon and mixed with 90 mL sterile bacteriological peptone water (NaCl, Merck, 8.5 g/L; Bacteriological peptone, Oxoid, Hampshire, UK, 1 g/L) in a stomacher bag and stored at 4 °C for 1 h. All the samples were then put in the ultrasonic bath (Millipore, Burlington, MA, USA) for 5 min, followed by a homogenization process in a stomacher (Seward, Stomacher 400, Worthing, UK) at high speed for 2 min. The samples were serially diluted, spread on plates by duplicates and incubated on Tryptic soy agar (TSA, Sigma-Aldrich, St. Louis, MO, USA) at 30 °C for 72 h for total aerobic count; on Rogosa agar anaerobically at 38 °C for 72 h for lactobacilli count; on Violet red bile dextrose agar (VRBD, (Merck KGaA, Darmstadt, Germany) for 24 h for the enumeration of *Enterobacteriaceae*; and on Malt agar (Merck KGaA) for 7 days at 23 °C for the enumeration of fungi. Two colonies per plate were randomly picked from TSA and Rogosa agar and purified on TSA and MRS plates, respectively. After purification, single colonies were enriched in Tryptic Soy Broth (Sigma-Aldrich, USA) for isolates from TSA and in MRS broth (Sigma-Aldrich, USA) for isolates from Rogosa agar. For cell harvesting, cultures were centrifuged at 6000 rpm for 5 min and the pellets were re-suspended in 1 mL of Hogness media, then frozen and stored at −80 °C until analysis.

### 2.3. Tea Brewing and Sampling

A mass of 10 g of each tea was brewed with 300 mL of sterile tap water at 90 °C for 2 min, following the instructions from the manufacturer. After brewing, the tea was filtered and cooled down to room temperature before being spread on plates and colonies isolated as described in Section 2.2.

### 2.4. Heat Resistance Test

Bacterial isolates picked from Rogosa agar and surviving tea brewing were cultivated in broth, washed in saline, and subjected to sterilized, hot water at 70 °C or 90 °C for 1 min and then directly transferred to ice. Water at ambient temperature was used as reference. The cooled samples were then spread on MRS plates (Merck, Germany) and incubated as described in Section 2.2. The test was performed in triplicates.

### 2.5. Bacterial Identification

DNA extraction, Polymerase Chain Reaction (PCR), gel electrophoresis, Sanger sequencing of the 16S rRNA gene, and data treatment were performed on Pu-erh and oolong tea isolates according to Uhlig et al. [17]. In short, isolates were suspended in Milli-Q water (Millipore, Milli-Q water system, Molsheim, France), followed by bead beating on an Eppendorf Mixer (model 4532, Eppendorf, Hamburg, Germany), and the supernatant was used for PCR. Primers ENV1 (5′-AGAGTTTGATIITGGCTCAG-3′) and ENV2 (5′-CGG ITA CCT TGT TAC GAC TT-3′) (Eurofins Genomics, Ebersberg, Germany), were used with TopTaq DNA Polymerase (Qiagen, Venlo, The Netherlands) according to the manufacturers’ instructions, using the following temperature profile: 95 °C for 3 min, 30 cycles of 94 °C for 1 min, 50 °C for 45 s, 72 °C for 2 min, and at the end, 72 °C for 10 min. PCR products were confirmed by gel electrophoresis and subsequently sent for sequencing at Eurofins Genomics, Germany. The sequences were trimmed to between 590 and 788 bp depending on sequence quality and compared to type strains in the Ribosomal Database project (RDP) by Seqmatch [18].

Twelve bacterial isolates surviving tea brewing were analyzed using Random Amplified Polymorphic DNA (RAPD) according to the method described by Quednau et al. [19]. DNA extraction and PCR followed the same procedure as described above, but with primer P73 (5′-ACG CGC CCT-3′) and the temperature programs as follows: 94 °C for 45 s, 30 °C for 120 s, and 72 °C for 60 s for four cycles, followed by 94 °C for 5 s, 36 °C for 30 s, and 72 °C for 30 s (with an extension of 1 s per cycle) for 26 cycles. The PCR session was terminated at 72 °C for 10 min, followed by cooling to 4 °C. The products were subsequently evaluated by gel electrophoresis.

### 2.6. Statistical Analysis

The colony count data were analyzed using SigmaPlot version 13.0 (SPSS Inc., Chicago, IL, USA). The differences between all groups were evaluated by Kruskal–Wallis one-way ANOVA on ranks. Differences between two experimental groups were assessed by Mann–Whitney U test, and results of *p* ≤ 0.05 were considered statistically significant.

## 3. Results

### 3.1. Viable Count

The total aerobic count from unbrewed tea leaves was highest in Pu-erh, ranging from 4.7 to 7.9 log CFU/g (Table 1) (*p* ≤ 0.01). The lactobacilli count in Pu-erh ranged from 5.1 to 6.4 log CFU/g. Both categories of bacteria were reduced to between 1.6 and 3.9 log CFU/g after brewing.

The total aerobic plate count of unbrewed leaves of Tie Guan Yin tea, Jingzhi, and Liu Bao Cha ranged between 2.9 and 3.8 log CFU/g, but after brewing, the counts were below the detection limit.

The total aerobic plate count in Nóng Xiang samples was below the detection limit of 1 log CFU/g. The same was valid for fungi counted on malt agar, and on VRBD plates for the enumeration of *Enterobacteriaceae* (detection limit 2 log CFU/g).

### 3.2. Heat Resistance Test

The pure cultures of *Heyndrickxia coagulans* that were subjected to heat stress at 70 °C in vegetive state, were reduced from around 7 log CFU/mL to between 1.5 and 3.8 log CFU/mL (Table 2). At 90 °C, the reduction went below the detection limit (1 log CFU/mL).

### 3.3. Sanger Sequencing

Pure cultures isolated from unbrewed leaves of Pu-erh and Tie Guan Yin sequenced by their 16S rRNA gene showed that the live bacteria isolated from the general-purpose medium TSA were primarily different species of the families *Bacillaceae* and *Paenibacillaceae* (Table 3). *Staphylococcus* spp. and *Micrococcus* spp. were also identified on the leaves. In the tea brew, different species of the families *Bacillaceae* and *Paenibacillaceae* were identified, and one isolate of an unidentified species of *Staphylococcus*.

On Rogosa agar, sequencing results generated exclusively *Heyndrickxia coagulans* both in the unbrewed leaves and in the brewed samples of Pu-erh.

### 3.4. Random Amplified Polymorphic DNA (RAPD)

As all bacteria surviving tea brewing on Rogosa agar were *H. coagulans*, those isolates were subjected to RAPD analysis to distinguish unique strains. Nine unique RAPD patterns were found in 12 isolates (Figure 1). Samples BC3 and BC4, BC9 and BC10, and BC11 and BC12 exhibit the same patterns.

## 4. Discussion

This study examined the culturable fraction of the microbiota, offering insights into the live microorganisms present in tea both before and after brewing, with a specific focus on species that may have implications for health and safety.

High total aerobic plate count was found for almost all investigated teas, but it was particularly high in Pu-erh teas (5.3–7.0 log CFU/g). Little is reported in the literature about viable bacteria in tea, but Hutková et al. [21] found viable count below 2.1 log CFU/g for Pu-erh. Similarly, the Liu Bao Cha tea in this study was also post-fermented, but had significantly lower viable counts. This indicates that the number of live bacteria on dark tea may vary within wide limits, which is probably due to conditions during post-fermentation. Hutková et al. [21] also found a higher level of *Enterobacteriaceae* in Pu-erh tea (2.03 log CFU/g), while in this study, the *Enterobacteriaceae* count was below the detection limit (1 log CFU/g) for all teas. Dry storage is essential when it comes to inhibition of *Enterobacteriaceae*, which are generally inhibited by a water activity lower than 0.94 [22].

In the present study, Nóng Xiang had a viable count below the detection limit and contained considerably less microorganisms than both Tie Guan Jin and Jingzhi. It could be hypothesized that the higher degree of enzymatic oxidation in this tea raises the concentrations of secondary polyphenols such as theaflavins and thearubigins. These compounds may act synergistically to inhibit microorganism growth [23]. Low levels of bacteria in black tea (0.7 log CFU/g) were also found by Hutková et al. [21].

The concentration of viable fungi was under the detection limit (1 log CFU/g) in all samples, which is surprising, especially for Pu-erh tea, as its production involves fermentation where *Aspergillus niger* has been reported to dominate [24]. One explanation could be that *Bacillus* spp. exhibits antagonistic properties against fungi—an example is *B. subtilis* that inhibits the growth of *A. niger* [23]. These findings highlight the role of microbial diversity and natural antagonistic interactions in shaping the microbiota of tea products, which contributes to their microbial safety.

No internationally agreed official microbiological limits are available for tea due to its long history of safe use, low moisture content, and the high content of antimicrobial substances. However, the European Tea Committee (ETC) and European Herbal Infusions Association (EHIA) have compiled a specification to facilitate trade in tea and promote a high-quality policy. It states that the limits for total aerobic count should be less than 7 log CFU/g and less than 5 log CFU/g for fungi in unbrewed tea leaves. Post-fermented teas, such as Pu-erh, are excluded from this specification [25]. The total aerobic count and the fungi count in the present study were far below those limits, which indicates good microbiological quality on the tested tea varieties.

Little is said in the literature about native, viable microorganisms in tea brew. Fernando et al. [26] found that the bacterial viability was reduced by more than 99.6%, but substantial levels of both viable fungi and bacteria remained after brewing. In the present study, bacterial levels were reduced to below the detection limit in all types of tea except for Pu-erh, possibly because Pu-erh contained higher concentrations of endospores before brewing. A reduction of more than 99.7% was seen, except for Pu-erh 6, which had a reduction of only 90.0%, indicating a higher concentration of thermotolerant bacteria.

As expected from a dry and hostile environment, most identities in this study included species of the families *Bacillaceae* and *Paenibacillaceae* with the ability to form endospores that provide protection against most environmental stress, heat included. However, non-spore-forming *Staphylococcus epidermis* and *Staphylococcus warneri,* which are related to skin flora were also found, indicating a certain level of human contamination. Moreover, soil-dwelling, non-spore-forming *Micrococcus* species were also identified. In a previous study by Hutková et al. [21], plant-related Gram-negative species, such as *Hafnia alvei*, *Acinerobacter junii* and *Sphingomonas* spp., were also found in tea.

Two isolates from Pu-erh were putatively identified as *Bacillus anthracis*, and one was identified as *Bacillus cereus*; however, the genetic variations of *B. cereus*, *B. anthracis*, and *Bacillus thuringiensis* exist mainly on episomes rather than on the chromosome genes [27,28]. Therefore, it is impossible to distinguish between these species purely with 16S rRNA gene sequencing, and consequently, it is difficult to draw any conclusions on the importance of those findings. Another human pathogen, *Staphylococcus epidermidis*, was found in the present study, but not after brewing. These findings emphasize the importance of monitoring potential pathogenic microorganisms in tea, but the absence of viable pathogens after brewing underscores the safety of the final product. However, proper storage and handling practices are essential for maintaining this safety, as they reduce the risk of contamination by harmful microorganisms.

In addition to evaluating pathogenic microorganisms, the analysis also highlighted the presence of beneficial microbes. Interestingly, the only bacterial species sequenced from Rogosa medium was *Heyndrickxia coagulans*. Rogosa agar is selective for *Lactobacillaceae*, but *H. coagulans* (formerly known as *Lactobacillus sporogenes*) tolerate the high acetate concentration and low pH of the media. All six Pu-erh teas contained *H. coagulans*, and nine different strains were found. *H. coagulans* has previously been identified in Pu-erh [29,30], and strains of *H. coagulans* have been claimed to have probiotic characteristics [31,32]. *H. coagulans* is qualified to be safe for consumption by the European Food Safety Authority in the QPS list (Qualified Presumption of Safety) and is approved by the United States Food and Drug Administration as GRAS (Generally Recognized As Safe) [31,33,34]. The identification of *H. coagulans* in tea is particularly significant, as its presence may contribute to the potential health benefits of tea consumption. The method used in this study shows that *H. coagulans* is viable, which is crucial for its ability to act as a potential probiotic bacterium when consumed. Furthermore, the processes observed during brewing and subsequent culturing on agar plates mirror the conditions that *H. coagulans* may encounter in the human gastrointestinal tract when tea is consumed. During brewing, spores of *H. coagulans* can survive and potentially be activated by sublethal heat. Similarly, in the gut, activated spores are exposed to nutrients and other germinants, prompting germination and the transition to their vegetative state. This suggests that when tea containing *H. coagulans* is consumed, the bacterium’s spores may germinate in the gut, allowing them to colonize and exert their potential probiotic effects [35]. This ability to activate, germinate, and function as a probiotic underscores the relevance of *H. coagulans* as a beneficial component of the tea microbiota, highlighting the importance of isolating and characterizing viable bacteria to ensure both the safety and health-promoting qualities of tea.

## 5. Conclusions

In the present study, it was shown that viable bacteria are most abundant in the post-fermented Pu-erh tea, but they were also present in Tie Guan Yin and Jingzhi, teas with lower oxidation levels. In contrast, the concentration of viable bacteria in dark oolong tea, which undergoes higher oxidation, was below the detection limit, likely due to the antimicrobial properties of phenolic compounds formed during oxidation. Notably, bacteria in Pu-erh tea were able to survive the brewing process to a certain extent, and nine distinct strains of *Heyndrickxia coagulans* were found to remain viable across different Pu-erh tea brews. This species is recognized for its probiotic potential and is considered safe for consumption by various national regulatory authorities. These findings underscore the importance of tea as a potential vehicle for probiotic delivery and highlight the relevance of microbial survivability during tea preparation.

## Figures and Tables

**Figure 1 microorganisms-13-00964-f001:**
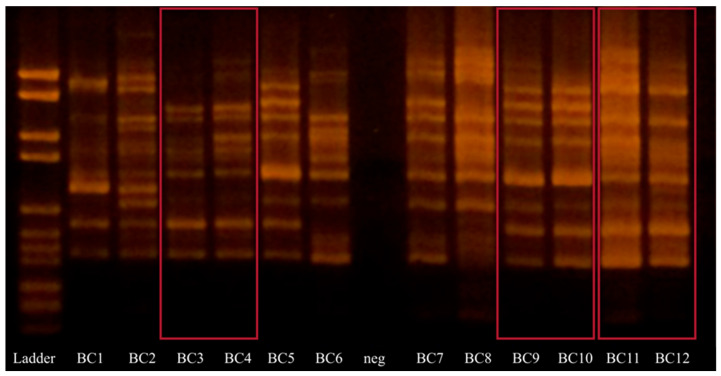
DNA bands of RAPD products from bacterial isolates extracted from tea brew. All are identified as *Heyndrickxia coagulans*. Neg stands for negative control. Identical DNA band patterns are marked in red.

**Table 1 microorganisms-13-00964-t001:** Total aerobic count and lactobacilli count of unbrewed tea leaves and tea brew. Counts are expressed as median log CFU/g leaves or mL tea.

	Total Aerobic Plate Count (Log CFU/g)		Lactobacilli Count (Log CFU/g)	
	Unbrewed Leaves	Tea Brew	Unbrewed Leaves	Tea Brew
Pu-erh 1	5.7 (5.4–5.8)	2.9 (2.8–3.0) **	5.6 (5.0–5.8)	2.6 (2.2–2.7) **
Pu-erh 2	5.4 (5.3–5.5)	2.8 (2.6–3.0) **	5.1 (4.9–5.2)	1.6 (1.6–1.8) **
Pu-erh 3	7.0 (6.9–7.1)	3.9 (3.6–4.0) **	6.4 (6.3–6.5)	3.6 (3.5–3.7) **
Pu-erh 4	5.3 (5.0–5.7)	2.4 (2.3–2.6) **	5.3 (4.9–5.7)	2.2 (1.9–2.6) **
Pu-erh 5	6.0 (5.7–6.0)	3.2 (2.8–3.4) **	5.9 (5.8–6.0)	3.3 (2.8–3.2) **
Pu-erh 6	4.7 (4.6–4.9)	3.7 (3.5–3.9) **	4.9 (4.5–5.2)	2.2 (1.0–3.6) **
Liu Bao Cha	3.8 (3.6–3.9)	<1	<1	<1
Tie Guan Yin 1	2.9 (2.7–3.0)	<1	<1	<1
Tie Guan Yin 2	3.6 (3.3–3.8)	<1	<1	<1
Jingzhi 1	2.9 (2.7–3.1)	<1	<1	<1
Jingzhi 2	3.5 (3.4–3.5)	<1	<1	<1

** indicates *p* ≤ 0.01 compared to unbrewed leaves in the same row and on the same growth medium.

**Table 2 microorganisms-13-00964-t002:** Viable count of *Heyndrickxia coagulans* isolates subjected to heat resistance test at 70 °C for 1 min. Counts are expressed as average of three replicates in log CFU/mL.

Treatment	Water at 4 °C Control	Water at 70 °C
*H. coagulans* 1	7.1	2.5
*H. coagulans* 2	7.1	2.7
*H. coagulans* 3	6.7	2.7
*H. coagulans* 4	7.0	1.5
*H. coagulans* 5	7.0	2.6
*H. coagulans* 6	6.8	3.8
*H. coagulans* 7	6.9	2.6
*H. coagulans* 8	7.1	2.1
*H. coagulans* 9	7.0	2.2
*H. coagulans* 10	7.0	2.3
*H. coagulans* 11	7.2	1.5
*H. coagulans* 12	7.0	2.9
*H. coagulans* 13	7.1	2.6

**Table 3 microorganisms-13-00964-t003:** Sanger sequencing results (16S rRNA) of viable bacteria from Pu-erh and green tea, and isolated from Tryptic Soy Agar (TSA).

	Unbrewed Leaves		Tea Brew	
	Tryptic Soy Agar (TSA)	Similarity (%)	Tryptic Soy Agar (TSA)	Similarity (%)
Pu-erh 1	*Paenibacillus cineris*	99.7	*Virgibacillus halophilus*	99.7
	*Lederbergia ruris*	99.4	*Brevibacillus parabrevis*	100.0
	*Lederbergia galactosidilytica*	97.3	*Bacillus thermoamylovorans*	99.1
Pu-erh 2	*Paenibacillus barengoltzii*	100.0	*Robertmurraya siralis*	99.7
	*Bacillus thermoamylovorans*	96.4	*Bacillus thermoamylovorans*	96.4
			*Paenibacillus barengoltzii*	100.0
Pu-erh 3	*Sphingobacterium bambusae*	94.9	*Virgibacillus halophilus*	99.4
	*Bacillus subtilis*	99.6	*Bacillus anthracis* **	100.0
	*Staphylococcus epidermidis* *	99.7	*Bacillus subtilis*	99.6
Pu-erh 4	*(Bacillus fortis) Brevibacillus fortis*	99.7	*Bacillus thermoamylovorans*	99.1
	*Bacillus sporothermodurans*	99.1	*Bacillus ruris*	100.0
	*Bacillus subtilis*	99.7	*Heyndrickxia oleronia*	99.7
	*Micrococcus yunnanensis*	98.8		
	*Staphylococcus epidermidis* *	99.1		
	*Virgibacillus halophilus*	99.7		
Pu-erh 5	*Bacillus farraginis*	100.0	*Bacillus shackletonii*	99.5
	*Micrococcus luteus*	99.1	*Bacillus anthracis*	100.0
	*Staphylococcus epidermidis* *	99.7	*Bacillus sonorensis*	99.7
Pu-erh 6	*Pseudomonas fluorescens*	100.0	*Paenibacillus pueri*	99.5
	*Staphylococcus warneri*	100.0	*Staphylococcus capitis*	100.0
	*Bacillus cereus* *	100.0	*Bacillus thermoamylovorans*	99.3
	*Bacillus subtilis*	100.0	*Paenibacillus barengoltzii*	100.0
	*Heyndrickxia oleronia*	99.5	*Virgibacillus halophilus*	99.2
	*Ornithinibacillus bavariensis*	100.0	*Bacillus ruris*	100.0
			*Heyndrickxia oleronia*	100.0
			*Bacillus aquimaris*	99.9
			*Bacillus shackletonii*	99.6
Tie Guan Yin 1	*Bacillus thuringiensis*	100.0	No colonies	
	*Staphylococcus warneri*	100.0		
	*Bacillus safensis*	100.0		
	*Staphylococcus capitis*	99.6		
	*Lysinibacillus sphaericus*	99.3		
Tie Guan Yin 2	*Paenibacillus polymyxa*	100.0	No colonies	
	*Heyndrickxia coagulans*	99.5		
	*Bacillus pumilus*	99.3		
	*Lysinibacillus sphaericus*	99.7		
	*Bacillus subtilis*	100.0		

* Denotes risk group 2 according to The List of Prokaryotic names with Standing in Nomenclature (LPSN) [20]. ** Denotes risk group 3 according to LSPN [20].

## Data Availability

The original contributions presented in this study are included in the article. Further inquiries can be directed to the corresponding author.

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
