# Peer review of "Viable and Heat-Resistant Microbiota with Probiotic Potential in Fermented and Non-Fermented Tea Leaves and Brews"

_microorganisms, 2025, doi:10.3390/microorganisms13050964_

Round 1
Reviewer 1 Report
Comments and Suggestions for Authors
The manuscript entitled "Viable and Heat-Resistant Microbiota with Probiotic Potential in Fermented Tea Leaves and Brews" is an interesting contribution to the microbiota description of tea, the introduction is well written, objective is clear, material and methods are reproducible, conclusions are supported by data, only minor details are needed to be clarified before accepting the manuscript
Comments
Title
The title emphasized viable and heat resistant, however the methods described more viable microorganisms in different tea types (leaves and brews), thus, the heat resistant microorganism was interesting, but they were observed in several types of tea, please reconsider it.
Material and methods
Sample collection
Please add more details about sample collection (i.e year of collect), also include particle size could be useful
Reviewer 2 Report
Comments and Suggestions for Authors
Abstract.
- Authors should structure the Abstract. Add Background, Methods, Results and Conclusion section.
Introduction
- The relevance of the stated problem for the world society should be expanded.
- Authors should update the references they used, especially 9-13, which are more than ten years old.
- The aim should be written more clear.
Materials and methods
- Authors used only 1 or 2 batches of each tea type in the study. Is it enough for obtaining valid and statistically significant results?
References
- The references should be updated as much as possible, there are many articles which were published more than ten years ago.
